# Sodium Content in Commonly Consumed Foods and Its Contribution to the Daily Intake

**DOI:** 10.3390/nu12010034

**Published:** 2019-12-21

**Authors:** Sonia Rosario Calliope, Norma Cristina Samman

**Affiliations:** Facultad de Ingeniería, Universidad Nacional de Jujuy, CIITED-CONICET, Ítalo Palanca Nº 10, Jujuy PC 4600, Argentina; soniroscal@gmail.com

**Keywords:** sodium content, street food, fast food, artisanal food, monitoring

## Abstract

Salt consumption in many countries of the world exceeds the level recommended by WHO (5 g/day), which is associated with negative effects on health. Public health strategies to achieve the WHO’s objectives include salt content monitoring, improved nutritional labelling and product reformulation. This study aimed to determine the sodium content in street food (SF), fast foods (FF) and artisanal foods (AF) of the Northwest of Argentina, which is not regulated. Moisture, ash and sodium were determined according to the Official Methods of Analysis (AOAC) in 189 samples from each of the three categories. The average and range values were: SF 520 (R: 74-932); FF 599 (R: 371-1093) and AF 575 (R: 152-1373) mg Na/100 g. Thus, general sodium content is high, which means that the consumption of a serving from most of the studied foods leads to an individual exceeding the recommended daily intake values. This study contributes to the knowledge of sodium content in evaluated foods and its contribution to the population intake. This reinforces the importance of implementing new public policies and regulations, advising consumers to check food nutritional labels andselect foods lower in salt content, raising food manufacturers’ awarenessabout the importance of reducing sodium content in foods they produce and in public health.

## 1. Introduction

Dietary salt intake has increased continuously in recent decades to reach 18 g/person/day in some regions of the world. This is associated with a marked increase in the prevalence of hypertension [1,2]. Batuman (2013) suggests that excess salt consumption may have detrimental effects on cardiovascular health regardless of hypertension [3]. More recent studies in humans confirmed that the low-salt diet lowers blood pressure in both normotensive and hypertensive individuals [4]. A well-known dietary trial on Dietary Approaches to Stop Hypertension (DASH) showed that reducing salt intake from 8 to 4 g/person/day reduced blood pressure among hypertensive and normotensive individuals [5]. There is currently a consensus that the beneficial effect of salt reduction begins with daily intake levels of 5 g/day or less [6,7]. Evidence also suggests that to obtain maximum benefit, salt reduction should begin at an early age [8,9].

In Latin America and the Caribbean, hypertension rates are between 20–40% and are among the highest in the world [10]. Worldwide, hypertension is the cause of 62% of stroke and 49% of coronary heart disease [11]. The National Survey on Risk Factors of the Ministry of Health of Argentina [12] shows that 34.1% of the adult population is hypertensive and it is estimated that the average salt intake per individual is around 12 g/p/day [13]; 37% of cardiovascular deaths are attributable to hypertension [14].

As a prevention system, the WHO and other health authorities recommend reducing salt intake by 5–6 g/day. Most countries have committed to the overall objectives of reducing salt in their food. The implementation of sodium reduction policies has been shown to be cost-effective and efficient in reducing overall cardiovascular diseases (CVD) by improving the health of the general population [15].

Argentina has established voluntary agreements based on the Healthy Argentina Plan of the Ministry of Health and has implemented the project titled “*Less Salt More Life*” since 2011, to reduce the use of salt in industrialized products and in artisan bakeries. Then, Law 26,905 [16] on the regulation of sodium consumption was enacted, which established maximum sodium contents for meat products and derivatives; farinaceous; soups, dressings and preserves. This law proposed the progressive reduction of salt content in industrialized foods to reach maximum values established for each group; besides, it also regulates the inclusion of warnings on the risks of excessive salt consumption in the packaging, promotes the elimination of salt shakers at restaurant tables, establishes that the salt content of each individual envelope should not exceed 500 mg and implements penalties for offenders of these regulations.

Estimates from the Ministry of Health of Argentina indicated that 65–70% of sodium intake in the diet comes from processed foods [17]. Technical feasibility is often cited by the industry as an impediment to reduce salt content, being necessary to develop new processes and technologies to deliver low salt content products. The food industry has an increasingly important role in public health. In order to verify whether the industry has reached the technical limits of salt reduction in the world, a study was carried out on the fulfilment of the sodium reduction targets in industrialized foods of the priority groups [18].

In addition, consumers purchase food outside their home; “*street*” sales throughout Latin America constitute a relevant phenomenon at the sociocultural, economic and health level.

On the other hand, artisanal foods contribute to the gastronomic richness of a country and represent the consumption of local foods; fast foods are generally associated with consumption close to work environments and young people. The consumption of these types of products is very common in Latin American countries, and is higher in populations with increasing socioeconomic and working conditions. All the above-mentioned food categories are unregulated products that depend on the practices and customs of the processors, which contribute to the lack of knowledge of sodium consumption levels. Food practices are a consequence of multiple individual, sociocultural and economic dimensions and of the accumulated learning by different generations [19]. Knowing about the perception that people have in regard to their daily salt consumption can be useful for the development of interventions aimed at reducing intake.

In order to continue with the research on foods that can contribute to the increase in sodium intake and consequently to non-communicable diseases, the objective of this study was to determine the sodium content in street foods, fast foods and artisanal foods that are consumed with high frequency in the Northwest of Argentinaand estimate their contribution to daily intake.

## 2. Materials and Methods 

### 2.1. Methodological Design 

A descriptive study was conducted through an anonymous survey, randomized to adults over the age of 18, who were in public places. They were personal surveys and were conducted by previously trained university students. The survey was designed with 10 simple-answer questions: yes, no, you donot know, there is no answer.

Descriptive statistics were used for the analysis of responses, which were also described through absolute and percentage frequencies.

To calculate final sample size (*n′*) the Cochran method was used [20], Formulas (1) and (2):(1)$$n_{0\;}=\;\frac{z^{2}\times p\times q}{e^{2}}$$
(2)$$n^{\prime }=\;\frac{n_{0}}{1+\;\frac{\left(n_{0\;}-1\right)}{N}}$$
where
*n*_0_: theoretical sample size;*n′*: final sample size;*z*^2^: zeta score is 1.96 for a 95% confidence level;*p*: maximum variability of the population at 50%. i.e., (0.5);*q*: 1−*p* = 0.5;*e*^2^: sampling error (0.05);1−*α*: 95% confidence level;N: population size (3,800,000), which covers three provinces of Northwestern Argentina.The resulting (*n′*) was 384 people.

A survey was carried out in order to determine the frequency of consumption of three food categories and which were the most consumed.

### 2.2. Food Categories 

Food categories studied: Street food (SF), fast food (FF), and artisanal food (AF).

The characteristics of each food category as taken for this work were following: 

Street Food: Ready-to-eat foods and drinks that sellers prepare and sell in public places. The stalls are located outdoors or under a roof, are easily accessible for the consumers and have and affordable prices.

Fast food: Are sold in formal food services such as cafeteria, take-out food and chain restaurants. In each place there is little variety of products and there is no delay in the service.

Artisanal food: The elaboration of artisanal foods is mostly in manual form and the production is carried out by small businesses or family groups. In them, the addition of ingredients depends on the criteria of the processor and the knowledge that was transmitted from generation to generation

### 2.3. Survey

The survey (Appendix A) was conducted primarily to collect information on types of food consumed and the frequency of consumption for each category mentioned (fast food, artisanal food and street food) and then select the subcategories (The most consumed types), which would be sampled and analyzed.

The survey was applied according to the following inclusion criteria:

A total of 486 men and women who are in public space, in fast food places and places of sale of artisanal food.

Ages were >18 years old (age range in which consumers can provide reliable information and deepen their preferences and motivations in relation to their diet).

They were asked whether or not they consumedfoods from the three defined categories. If an affirmative answer was given, they informed about the frequency of consumption and listed the 3 most consumed products for each category. With this information, a list from highest to lowest was prepared with foods classified by frequency of occurrence and the first seven of each category were included for the study.In the survey it was also asked the minimum serving size of each food (MSS) per meal.

### 2.4. Food Sampling Plan

7 food items from each category were selected and 9 samples from each were analyzed.7 foods × 3 categories = 21 foods to sample21 foods × 9 samples/food = 189 samples to collect (total)

The moisture determination was made in triplicate (567 determinations), ashes and sodium were determined in duplicate (378 determinations of each).

As a sample unit size for each food, 300 g or more were set. Each sample unit was the one corresponding to a plate or serving (as it is usually consumed). For the food of smaller portions, aminimum quantity of units was included until completing the fixed size.

The sampling was carried out in 3 provinces of Northwestern Argentina: Jujuy, Salta and Tucumán, as they are places where the consumption of the selected food categories is widespread.

### 2.5. Sample Preparation

The acquired samples were packed in hermetically sealed polyethylene bags and kept refrigerated, labelled with food name, place of sampling, date, time and responsible person and after they were transferred to the laboratory. The moisture of each sample was immediately determined and the rest was dried, homogenized and kept packaged and refrigerated until processing. Finally, by quartering the sample size was reduced to analyze moisture, ashes and sodium. A description of the ingredients of each food was made and photographs were taken to create a database.

### 2.6. Chemical Analysis

Association of Official Analytical Chemists (AOAC) methodswere performed [21].

Moisture: (Method N° 945.15); ash: by calcination in muffle (Method N° 923.03); sodium: by suspension of ashes in 3N nitric acid and reading in flame photometer model ZF 250/240 (Method N° 963.15).

### 2.7. Statistical Analysis

The information collected in the surveys was analyzed on qualitative variables, using the deductive method to describe the most common trends. Quantitative values were expressed as mean ±SD. Variance analysis between categories was performed and Tukey’s test at a significance level of 0.05 was used.

Data analysis was performed using InfoStat-Statistical Software version 2008, and a *p*-value < 0.05 was considered to indicate statistical significance.

## 3. Results

The number of surveys carried out was 486. Table 1 reports the descriptive statistics and the percentages of the survey responses. Of the surveys, 131 (27%) were answered by men (19 to 47 years old) and 355 (73%) by women (20 to 40 years old). More than 70% of respondents showed their concern and interest in healthy eating, however very few of them reported knowing the daily recommended sodium intake. Only 12% admitted having cardiovascular disease. A high percentage responded knowing the difference between salt and sodium. According to the answers obtained, the seven most named foods were selected for each proposed category (street food, fast food and artisanal food) to be analyzed. Figure 1 shows the distribution of all the foods mentioned in each category. It can be seen that some foods were named in more than one category; the highest frequency was selected for its definitive classification. Figure 2 shows the responses on frequency of consumption including the three categories studied. It was adopted to perform some calculations (not shown in this manuscript) 4–6 times per week.

Table 2 shows the three food categories under study with the respective subcategories selected, the size of each sample unit, usual portion size, description of each product and a photograph. These products are generally classified as very palatable and easily acquired, especially street food. They are also cheaper than those acquired in formal places. Often more than one usual serving is consumed.

Figure 3a–c shows the average contents, maximum and minimum values by category/subcategory of food. As it can be observed, there is great variability for each subcategory. If the sodium contents of the total foods analyzed are compared with those established by traffic light, it can be observed that only French fries can be classified as low sodium (≤120 mg Na). Among the medium-content foods (>120 and <600 mg of sodium), there are four foods from the fast food category, five artisanal foods and three of street-sale food. All other foods analyzed are classified as high in sodium (≥600 mg Na/100 g).

Figure 4a–c shows the contribution of sodium amount contained in a commonly consumed portion of each food (per subcategory), calculated with the average sodium content value. The WHO recommended daily intake value was included.

These results show that sodium intake depends so much on the processor, due to the amount of salt that each of them adds and also on the consumer when choosing the portion size.

## 4. Discussion

In Argentina, projects have been carried out to reduce salt consumption since 2011 by the national government. They worked on the project “*less salt more life*” and healthy school kiosks. In 2013, a law on the regulation of sodium content in industrialized foods was enacted. However, the average salt consumption continues to exceed 12 g salt/day. These projects and legislation do not consider the type of food studied, which due to its high consumption and salt content can be important contributors to the daily intake. The results of this study confirm that hypothesis. Therefore, it is of special interest that these foods are also taken into account to develop public policies related to sodium reduction. This study allowed us to know that consumers do not consider the size when selecting a portion of food or that the related contribution may be excessive. In general, they also do not know the foods with the highest salt intake or the WHO intake recommendation.

The variability observed in the salt content between products of the same subcategory allows us to conclude that sodium content depends primarily on the processor. In some foods, like loin sandwich, which is easily manufactured, sodium content is due to the ingredients used. In others, e.g., French fries, the addition of salt depends a lot on the food habits of the consumer.

The acquisition of food outside home in Latin America is already a culturally established activity, so it is a priority to constitute actions that also control this type of food. The results of this work show that the three categories studied are of habitual consumption and the contribution to sodium intake is high and can have detrimental effects on health. Since sodium reduction is a public health objective, Argentina should promote the implementation of sodium reduction policies in these categories, expanding the list of foods promulgated by law and not limited only to industrial foods. This study would offer criteria for classifying foods in these categories according to their critical nutrient content.

## 5. Conclusions

Due to the low knowledge of consumers, more information campaigns should be carried out on the recommended daily amount of salt and on the risks of consuming it in excess.

The sodium content in the three categories of food studied is high and the variability in each subcategory is wide. The high variability indicates that the greatest influence on the sodium content in this type of product is the salt added by the processor. The contribution of this type of food to the daily intake can be very high, for the high frequency of consumption informed in the surveys conducted to consumers (4 to 6 times/week and in opportunities more than once per day). This shows that it is necessary to raise awareness among food processors about the importance for health of reducing sodium consumption and the relevance they have to achieve this goal.

Based on the results obtained, it is observed that the sodium intake of the Argentine population continues to be high, even though the efforts made by public health; therefore, an analysis is required on the types of food consumed daily. This study provides valuable information regarding the contribution of sodium to the diet by street food, artisanal food and fast food, which are of consumed in high quantities. Several of these foods would exceed the recommended maximum daily sodium intake of the WHO with just a single portion. This would allow the formulation of new public policies to implement regulations and to keep the consumer informed about the possible risks of consuming them.

Considering Joint Resolution 12/2019 of the Ministry of Health Regulation and Management and the Secretary of Food and Bioeconomy of the Nation (which modifies article 21 of the Argentine Food Code [22] and establishes that any person who may be in contact with food must complete and approve a training course on safe food-handling), and considering that the World Health Organization declares that it is essential to prioritize training of the food-handling staff, with an approach based on the prevention of foodborne diseases, the concept could be extended so that food-handlers should alsobe trained in basic aspects of food, nutrition and health.

## Figures and Tables

**Figure 1 nutrients-12-00034-f001:**
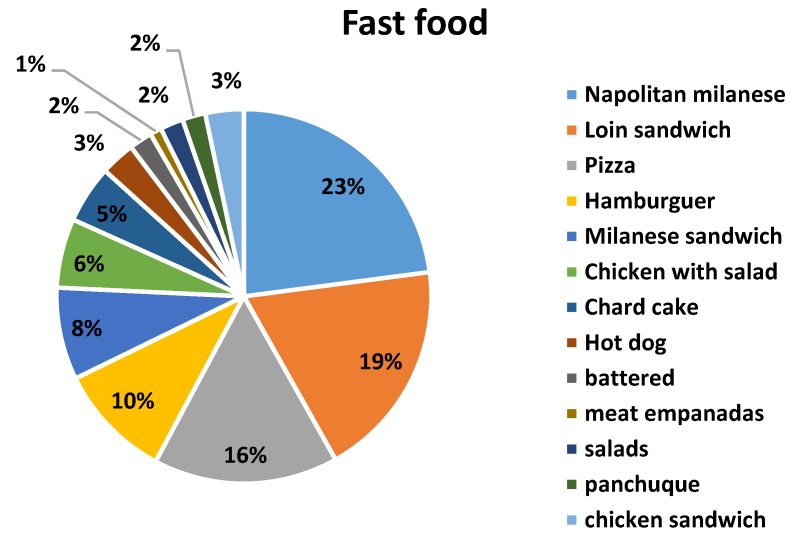
Distribution of total foods mentioned by consumers.

**Figure 2 nutrients-12-00034-f002:**
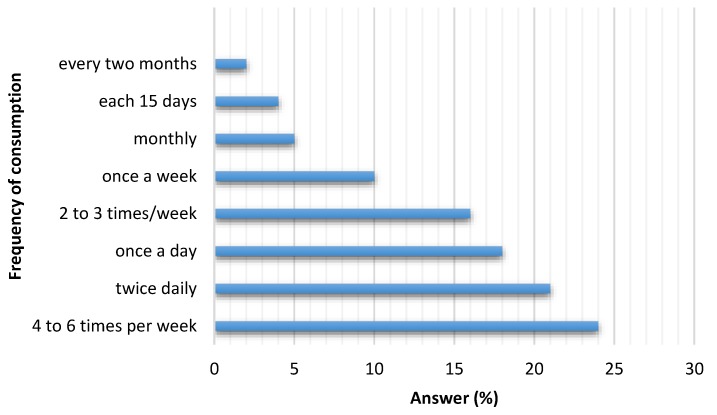
Frequency of consumption of these types of foods.

**Figure 3 nutrients-12-00034-f003:**
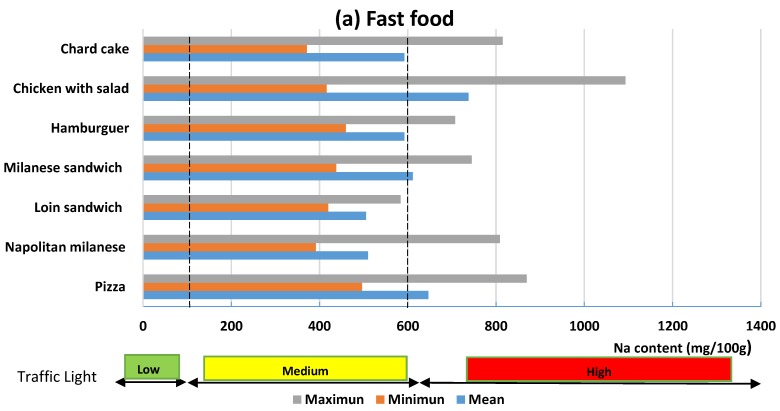
Classification of foods in low, medium and high sodium content (mg of Na/100g food).

**Figure 4 nutrients-12-00034-f004:**
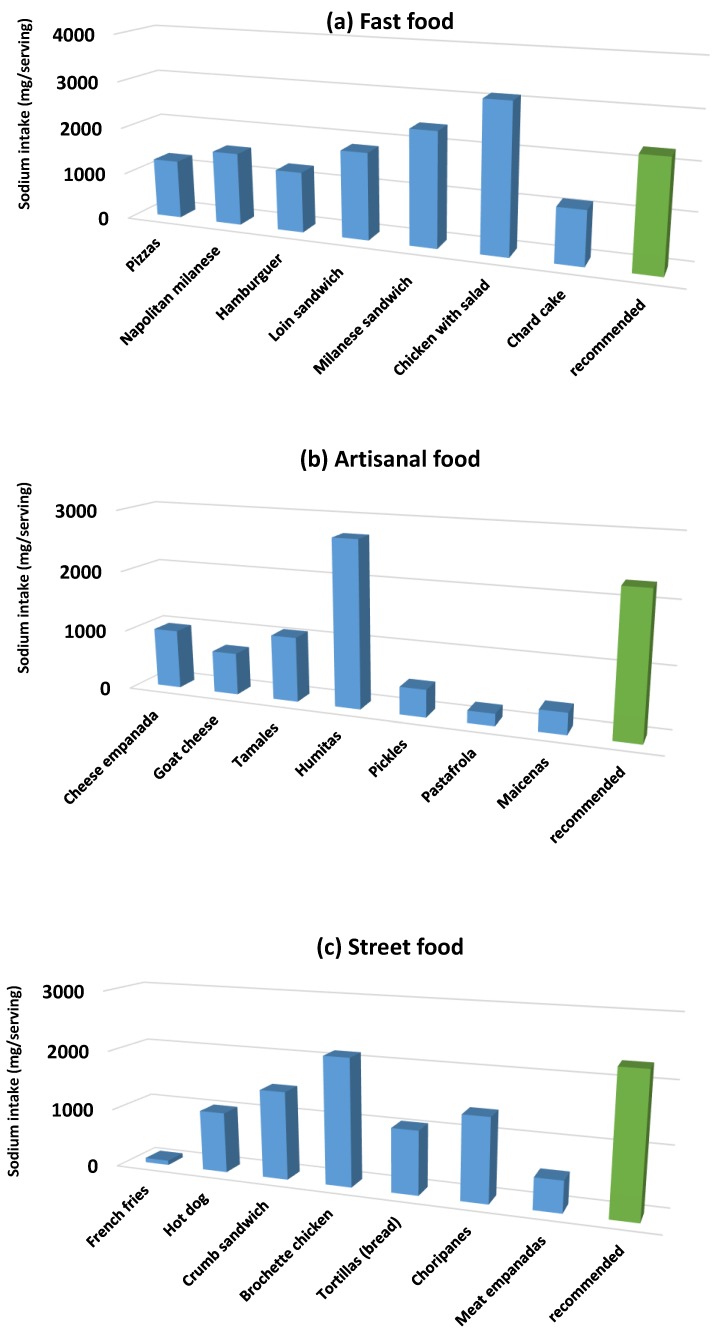
Average sodium intake per serving.

**Table 1 nutrients-12-00034-t001:** Descriptive statistics and percentages of the survey responses.

**Variable**	**Category**	**Frequency (%)**
Age	up to 20 years	15.00
	up to 30 years	70.00
	up to 40 years	8.33
	up to 50 years	6.67
Sex	Female	73.00
	Male	27.00
**Survey Questions**	**Answers**	**Frequency (%)**
Do you try to eat healthy diet?	Yes	76.79
	No	7.14
	I donot know	3.57
	Unanswered	12.50
Do you try to minimize salt intake?	Yes	73.21
	No	8.93
	I donot know	8.93
	Unanswered	8.93
Do you know which foods contain much salt?	Yes	66.07
	No	21.43
	I donot know	5.36
	Unanswered	7.14
How many times do you add salt on your meals?	Always	3.57
	Sometimes	41.07
	Rarely	33.93
	Never	21.43
How much salt do you think consume?	Too much	26.79
	Few	67.86
	I donot know	3.57
	No answer	1.79
Have you had hypertension problems?	No	87.50
	yes	12.50
Do you know if there is a recommended amount of salt intake?	Yes	17.86
	No	76.79
	I donot know	5.36
Do you know the difference between salt and sodium?	Yes	67.86
	No	17.86
	Unanswered	5.36
	I donot know	8.93

**Table 2 nutrients-12-00034-t002:** Categories and subcategories of selected foods.

Category	Subcategories	Ingredients/UW-MSS *	Photo
**Artisanal food**	Cheese empanada	Dough, cheese, onion, pea, egg, potato UW: 56 g MSS: 168 g (3 units)	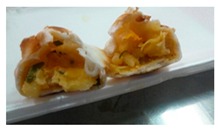
	Goat cheese	Goat milk, rennet, salt. UW: 366 g MSS: 60g	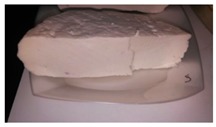
	Tamales	Cornmeal, meat, potatoes, condiments UW: 142 g MSS: 284 g (2 units)	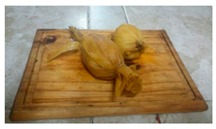
	Humitas	Ground fresh cornpaste, semi-hard cheese, pumpkin, seasonings UW: 231 g MSS:463 g (2 units)	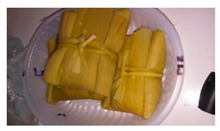
	Pickles	Chili pepper, carrot, onion, broccoli, cauliflower, vinegar, salt UW: 562 g MSS: 50 g (2 table spoon)	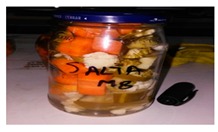
	Pastafrola	Wheat-flour, butter, eggs, quince jam, sugar UW: 405 g MSS: 102 g	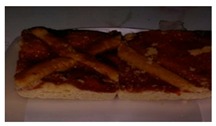
	Maicenas	Wheat flour, corn starch, eggs, butter, sugar, caramel, grated coconut UW: 53 g MSS: 159 g (3 units)	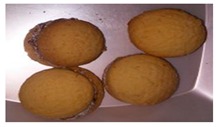
**Fast food**	Pizzas	Base dough: Wheat flour, oil, salt, yeast; toppings: Cheese, tomato sauce, olives, chili pepper UW: 513 g (1 medium pizza) MSS: 193 g (3/8 portions)	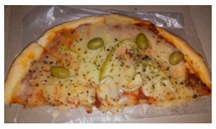
	Napolitan milanese	Breaded beef-steak (covered with egg and breadcrumbs); toppings: tomato sauce, cheese, cooked ham, olives; accompanied by fried egg and Frenchfries UW: 608 g (1 plate) MSS: 304 g (1/2 plate)	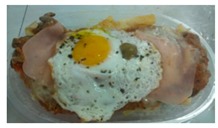
	Simple Hamburger	Bread, beef burger, egg, tomato, lettuce, ham, cheese, mayonnaise, ketchup UW: 217 g MSS: 217 g	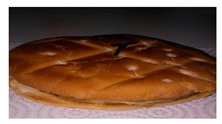
	Loin sandwich	Bread, beef-steak, fried egg, tomato, lettuce, mayonnaise UW: 364 g (1 sandwich): MSS: 364 g	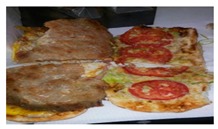
	Milanese sandwich	Bread, milanese, fried egg, ham, cheese, tomato, lettuce, mayonnaise UW: 394 g (1 sandwich) MSS: 394 g	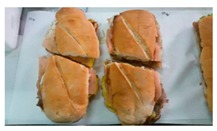
	Chicken with salad	Chicken leg and lettuce and tomato salad UW: 422 g (1 plate) MSS: 422 g	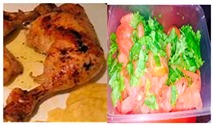
	Chard cake	Wheat-flour dough, chard, beaten egg, onion, grated cheese, condiments UW: 383 g MSS: 192 g (Half unit)	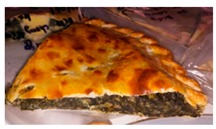
**Street food**	French fries	French fries, mayonnaise, salt UW: 81 g MSS: 81 g	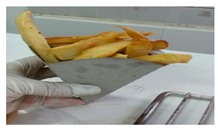
	Hot dog	Vienna bread, sausage, mustard, mayonnaise, ketchup UW: 77 g MSS: 154 g (2 units)	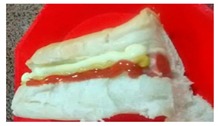
	Crumb sandwich	Crumb bread, ham, cheese, mayonnaise UW: 65 g MSS: 195 g (3 units)	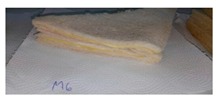
	Chicken brochette	Spiced chicken, sliced onion, pepper or roasted tomato. UW: 183 g MSS: 386 g (2 units)	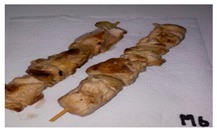
	Tortillas (bread)	Wheat flour dough, fat, water UW: 426 g MSS: 213 g (Half unit)	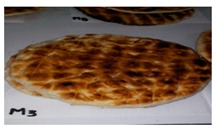
	Choripan	French-type bread and pork sausage, mayonnaise. UW: 179 g MSS: 179 g	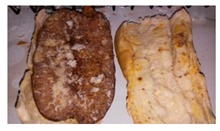
	Meat empanadas	Wheat flour dough, ground meat, onion, pea, potato, hardboiled egg UW: 59 g MSS: 177 g (3 units)	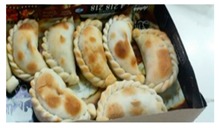

* UW: Unit Weight; MSS: Serving size of each food per meal.

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
