# Peer review of "Sodium Content in Commonly Consumed Foods and Its Contribution to the Daily Intake"

_nutrients, 2019, doi:10.3390/nu12010034_

Round 1
Reviewer 1 Report
Thank you to the authors for responding to my points. I did note that the sample size is stated as 384 on line 140 and then 486 on line 183 - please clarify.
Author Response
Point 1: Thank you to the authors for responding to my points. I did note that the sample size is stated as 384 on line 140 and then 486 on line 183 - please clarify.
Response: The representative minimum sample size (n0) is calculated using formula 1. Then an adjustment is made to the equality proposal to determine the actual sample size (n´) to be surveyed when the population value "N" is known (Formula 2). The calculated sample size (n´) was 384. In other words, at least 384 people should be surveyed. In the field work, 486 surveys were carried out, which improves the representativeness.
Reviewer 2 Report
This manuscript deals with sodium consumption in Argentina. It presents and analyses a survey done in Argentina concerning food consumption and sodium daily intake. The foods are categorised in 3 categories: street food, fast food and artisanal food. Their chemical analyses, including sodium content, are also presented. This is a valuable topic because, despite recommendations by healthy organisms such as WHO since many years to reduce salt consumption, the fixed objectives are far to be reached in most countries over the world. Thus, the excessive consumption of sodium salt remains an important public health problem.
Questions and comments
L81: Concerning the methodological design, for non-specialist of statistical survey analysis as me, the calculation is not very clear, in particular what represent n’ and no. I think a short explanation with few clear and simple sentences would be helpful for understanding of non-specialists. Is there a bibliographic reference for this statistical methodology?
In my mind, 486 people is rather low for such survey. Is it really representative of the overall considered population?
L99: The definition of the 3 categories of foods should be precisely defined. Here, it is not clear for example, what is called fast food: only what is consumed in fast food restaurants such as MacDonald or the same consumed at home; how street foods and artisanal foods are differentiated? It could be the same in term of food composition.
I do not really understand how the link is done between the survey responses in Table 1 and the sodium food content presented in the other table and figures. The questions are on general food habits, basic knowledge and healthy information without any medical data. There is no question concerning the diet of each participant and his or her sodium intake. Thus, it appear as two independent studies.
In table 2, the content of sodium and the average sodium intake of each food could be added to complete information.
Finally, are and how the frequency of consumption of people for each of the 3 food categories are taken into account?
Author Response
L81: Concerning the methodological design, for non-specialist of statistical survey analysis as me, the calculation is not very clear, in particular what represent n’ and n0. I think a short explanation with few clear and simple sentences would be helpful for understanding of non-specialists. Is there a bibliographic reference for this statistical methodology?
In my mind, 486 people is rather low for such survey. Is it really representative of the overall considered population?
The representative minimum sample size (n0) is calculated using formula 1. Then an adjustment is made to the equality proposal to determine the actual sample size (n´) to be surveyed when the population value "N" is known (Formula 2). The calculated sample size (n´) was 384. In other words, at least 384 people should be surveyed. In the field work, 486 surveys were carried out, which improves the representativeness.
Regarding the subject, the following bibliographical reference was added to the text.
Cochran, W. Sampling techniques. Third edition. John Wiley & Sons, 2007, 363-365.
L99: The definition of the 3 categories of foods should be precisely defined. Here, it is not clear for example, what is called fast food: only what is consumed in fast food restaurants such as MacDonald or the same consumed at home; how street foods and artisanal foods are differentiated? It could be the same in term of food composition.
The definitions of these food categories are not accurate. The classification is only to facilitate the study of sodium content in different type of foods no legislated.
Each category contains different foods; In addition, a food can be found in more than one category. The selection was made by taking each food in the category that was most mentioned by the consumers surveyed. For example, the empanadas were mentioned in the categories "street food" and "artisanal food", those of meat was included in the category street food and those of cheese in artisanal food because they received the largest number of responses in them, respectively.
All the samples were acquired outside home.
The characteristics of each food category as taken for this work were added to the manuscript (Line 104-111).
I do not really understand how the link is done between the survey responses in Table 1 and the sodium food content presented in the other table and figures. The questions are on general food habits, basic knowledge and healthy information without any medical data. There is no question concerning the diet of each participant and his or her sodium intake. Thus, it appears as two independent studies.
The survey was conducted primarily to collect information on types of food consumed and frequency of consumption for each category mentioned (Fast food, Artisanal food and Street food) and then select the subcategories (The most consumed types), which would be sampled and analyzed.
The complete survey was added. The survey additionally allowed having a general knowledge of the characteristics of the population surveyed and their knowledge regarding the importance of sodium consumption.
In table 2, the content of sodium and the average sodium intake of each food could be added to complete information.
The requested information is found in Figures 3 and 4. We consider not including them in Table 2 so as not to duplicate the information.
Finally, are and how the frequency of consumption of people for each of the 3 food categories are taken into account?
Figure 2 shows the responses on frequency of consumption of the three categories studied, given that a general question was asked:
How often do you consume these types of foods?
The frequency that was adopted to perform some calculations (not shown in this manuscript) was 4-6 times per week. It was commented in conclusion Line 248.
Round 2
Reviewer 2 Report
I agree with the answers and modifications.
This manuscript is a resubmission of an earlier submission. The following is a list of the peer review reports and author responses from that submission.
Round 1
Reviewer 1 Report
Thank you for the opportunity to review for Nutrients. The paper presents some interesting results on salt levels in commonly consumed foods in Argentina. It may be a useful study within the country. Some aspects require greater details.
Introduction
- Well written and discusses the context of the issue. Covers lots of areas as required.
- The link to hypertension is a fair point, but I cannot understand the discussion of examples using chimpanzees when there is a lot of evidence for impact on humans (the focus of your paper). Suggest revise/drop.
Materials and Methods
- How was the survey sampled? What were the sample characteristics? Was it representative? What are the implications if not? Lacks details on the design and implementation of the survey.
- Could have tidied up the presentation of text here.
- Not clear what was being tested in the statistical analysis. Suggest link to the aims of the project and select appropriate analyses here.
- Unclear how fast food and street food differed - hot dogs and french fries would cover both categories for example, but are in street food.
- The selection of 7 most common foods is arbitrary and doesn't feel scientific or justified.
Results
- I cannot see the results of the survey of public - all of the graphs are for salt levels within foods. They should be reported as basics (e.g. sample characteristics) to help understand the interpretation of the findings.
- Figure 2 - unsure what is being tested here and understanding the outputs of the statistical analyses in the graph. Needs clarifying.
- I am unclear what the three graphs are adding; they seem to be presenting the same message and results but slightly different. Feels repetitive.
Discussion and Conclusions
- The discussion lacks detail and depth to expand on the meaning, interpretation and importance of the findings. There is not link to the evidence base and how they fit in. Under-developed in comparison to the front end of the paper.
Reviewer 2 Report
This study investigated the sodium content of foods consumed outside home in Argentina and reported the necessity of reducing sodium content of the foods for public health. However, it is doubtful that this manuscript has significant meaning. The major concerns are listed as follows.
The study design seems to be rough. What is the reason to classify three categories of street food, artisanal food, and fast food? Each category contains diverse foods such as for snack, main meals, and side dishes. The dishes can be served separately or combination with others. In addition, the variability of sodium content in each subcategory shows the big difference by the processor. Is there any significance to compare means of three groups (Fig 2)? It would be reasonable to classify foods into snack, main meals, and side dishes. Considering the wide variability of sodium content in each food, Fig 3 shows only that most of food have considerable amount of sodium compared with recommended daily sodium intake level. Each bar graph does not give information more than that. Authors mentioned that they aimed to estimate the contribution of the foods to daily intake. The discussion needs to be further developed to make estimation, based on the frequency of foods and intake. In addition, previous studies or reports in other countries should be considered.